# Lonely, but Not Alone: Qualitative Study among Immigrant and Native-Born Adolescents

**DOI:** 10.3390/ijerph182111425

**Published:** 2021-10-30

**Authors:** Katrine Rich Madsen, Tine Tjørnhøj-Thomsen, Signe Smith Jervelund, Pamela Qualter, Bjørn E. Holstein

**Affiliations:** 1The National Institute of Public Health, University of Southern Denmark, 5230 Odense, Denmark; titt@sdu.dk; 2Section for Health Services Research, Department of Public Health, University of Copenhagen, 1014 Copenhagen, Denmark; ssj@sund.ku.dk; 3Manchester Institute of Education, University of Manchester, Manchester M13 9PL, UK; pamela.qualter@manchester.ac.uk

**Keywords:** loneliness, adolescence, qualitative methods, ethnicity, immigration

## Abstract

This paper explores loneliness as it is understood and experienced by adolescents, with a special focus on the importance of their migration status. We recruited students from five schools following a maximum variation sampling scheme, and we conducted 15 semi-structured, individual interviews with eighth-grade adolescents (aged 14–15 years) that were immigrants, descendants, and with a Danish majority background. A thematic analysis was applied with a special focus on differences and similarities in understanding and experiencing loneliness between adolescents with diverse migration status. The results showed more similarities than differences in loneliness. Generally, loneliness was described as an adverse feeling, varying in intensity and duration, and participants referenced distressing emotions. Feeling lonely was distinguished from being alone and characterized as an invisible social stigma. A variety of perceived social deficiencies were emphasized as causing loneliness, emerging in the interrelation between characteristics of the individual and their social context. The results add to the current literature by highlighting that it is not the presence of specific individual characteristics that causes loneliness; instead, loneliness is dependent on the social contexts the individual is embedded in. Differences across migration status were few and related to variations in the adolescents’ individual characteristics. The findings highlight the importance of (1) studying the characteristics of both the individual and the social context in research on the antecedents to adolescents’ loneliness, and (2) applying this perspective in other studies on the importance of migration status.

## 1. Introduction

Intense and prolonged feelings of loneliness during adolescence may have serious consequences for mental and physical health, behavior, and academic achievement in later life [1,2,3,4,5,6]. Consequently, insight into the nature of loneliness and its sources among adolescents is a public health concern. Due to growing migration and global mobility, adolescents with different ethnicities, languages, norms, and traditions constitute a growing part of today’s multiethnic societies [7,8]. Adolescents with an ethnic minority background (immigrants or descendants of immigrants) may experience acculturation challenges in terms of language barriers, discrimination, or loss of social relations due to migration [9,10,11], which may induce loneliness. Yet, research on whether and how adolescents’ migration status is related to loneliness is sparse [12]. This paper addresses this gap by exploring the nature of loneliness among adolescents and the role of their migration status.

### Loneliness in Adolescence

Loneliness seems most prevalent and intense during adolescence [13,14,15]. In this period, the desire to belong and feel accepted is particularly strong, and establishing and maintaining satisfying and reliable relations are important but challenging tasks [14,15], (pp. 269–290, [16]). In adolescence, social relations with peers gain increased significance, and friendships are expected to include a high degree of loyalty and intimacy. Higher expectations to one’s social relations and an increased focus on one’s social status may also contribute to loneliness during adolescence [14,15,17].

Loneliness has been defined in a variety of ways. Nevertheless, most definitions consider loneliness to be the result of perceived deficiencies in one’s social relations [16,18] and include “an affective tone described as painful, sad, or aching and a cognition of oneself as alone or isolated” [19]. According to several scholars (e.g., [14,16,20]), loneliness is a basic fact of life, and most adolescents may, therefore, feel lonely at some point. For most, the feeling is transient and related to a particular situation. For others, the feeling of loneliness can be prolonged and intense.

There are few qualitative studies exploring how loneliness is understood and experienced, and they were mainly conducted among 8–13 year old children and adolescents from the majority population [20,21,22,23,24]. In a US study, Kristensen [23] interviewed 8–10-year-olds about their experiences of loneliness. Nine themes (e.g., intense feelings, coping, physically alone) were identified as covering the experience of loneliness while being unhappily disconnected, as a unitary construct across the themes. An Australian study explored the conceptualization of loneliness in 9–11-year-olds by interviews. Most participants explained loneliness by referring to social deficits (e.g., no one to play with) and distressing emotions (e.g., sad, bored). A small proportion of children also made reference to self-attributions, e.g., being different [21]. Another Australian study explored the nature of loneliness, aloneness, and friendships among children and adolescents aged 10–15 years old. The authors found that loneliness was a multifactorial experience influenced by two main constructs: perceptions of aloneness, and not feeling connected with friends [22]. A Canadian study by Hymel and colleagues [20] included interviews with 8–13-year-olds and found that feeling lonely was a multidimensional phenomenon including three interrelated components: an affective dimension, a cognitive dimension, and a set of interpersonal situations or contexts. Although there are similarities in the findings of these few studies, there are also important differences, for instance, regarding the dimensionality of loneliness and perception of the origin and context of loneliness. Therefore, more qualitative studies are warranted.

Research on whether and how adolescents’ migration status is related to loneliness is sparse and mainly restricted to epidemiological studies. Adolescents who have migrated themselves (immigrants) or who are born into immigrant families (descendants) represent heterogeneous groups. Yet, they are similar in the sense that they are more likely to experience challenges in terms of language barriers [25,26], parent–adolescent acculturation gaps [27], less time with parents in the new country [27], perceived discrimination [25,28], cultural conflicts [25], and low self-esteem [9], which may all be related to loneliness. Moving away from friends or family is a common explanation for loneliness, making immigrants particularly vulnerable to feeling lonely [16,27,29,30]. The concept of “cultural loneliness” was suggested by van Staden and Coetzee [31] and covers the loneliness someone may feel when they are in a foreign culture where they do not feel understood and are unable to reciprocate matters about cultural meaning. The experience of loneliness may also differ across cultures because culture shapes the individual’s expectations about optimal size and quality of relations [31,32]. Qualitative research can provide insight into this area and elucidate how adolescents understand and experience loneliness with special attention paid to the role of their migration status and context they are embedded in [12,21], which is currently lacking [20,21,22,23,24]. There is, therefore, a need for qualitative studies, which focus on people’s experiences of a certain phenomenon, e.g., loneliness and context.

This study explores whether there are differences in understanding and experiencing loneliness based on migration status. Thus, we address the research gap in our understanding of how nativity and culture influence loneliness by interviewing 14–15 year old adolescents. The aim of the current study was, therefore, to explore how adolescents understand and experience loneliness in their everyday social lives with a special focus on the role of their migration status. To avoid stereotyping minority groups, the analysis focused on both differences and similarities across migration status, as differences between minority and majority groups are easily demonstrated even when similarities are more pronounced [33,34].

## 2. Materials and Methods

We followed the COREQ checklist by Tong et al. [35] to ensure explicit and comprehensive reporting of this qualitative study.

### 2.1. Recruitment, Data Collection, and Participants

Data stem from 15 semi-structured, individual interviews with 14–15 year old adolescents with diverse migration status (Danish origin, immigrants, and descendants) attending eighth grade at five public schools in Denmark. We applied maximum variation in the sampling strategy of schools, which resulted in a study population with mixed ethnic backgrounds attending schools with low, middle, and high proportions of ethnic minorities. Statistics Denmark provided information about the proportion of schoolchildren with an ethnic minority background (immigrants and descendants) based on the schoolchildren and their parents’ country of birth and citizenship. Five schools were strategically selected to include schools with low (schools 1 and 3, see Table 1), middle (school 2), and high (schools 4 and 5) proportions of ethnic minorities. All five schools agreed to participate.

Participants from each school were recruited through a process of three meetings with the first author (K.R.M.), who also conducted the interviews. The intention of these steps was to build a confident relationship with the adolescents before the interviews about loneliness. At the first meeting, K.R.M. visited the school classes during a lecture to give general information about herself and the purpose of the study. Prior to the visit, the teacher had distributed information about the study to parents with information about the study, i.e., that the study was voluntary, that data were treated confidentially, and that they had the opportunity to withdraw their child from the study. During this first visit, the adolescents answered a small questionnaire about themselves and their experiences with feeling lonely, left out, and not accepted. Those adolescents who were willing to participate in later interviews gave their consent by reporting their name and contact information. At four schools, the majority reported their name and contact information. At the fifth school, only two reported this information. At all five schools, a second meeting took place 1 week later. Here, focus group interviews were carried out with all adolescents who provided contact information at the first meeting, and who gave their consent to participate in later interviews. These interviews were part of K.R.M.’s PhD project, but not related to loneliness; therefore, they are not included in this paper. However, this second meeting was an important step in building a confident relationship between participants and K.R.M. At the third meeting, the adolescents were interviewed about their understanding and experiences of loneliness individually. Participants in these interviews were identified through purposive sampling on the basis of their answers to the questionnaire from the first meeting about country of birth and their experience with feeling loneliness, left out, and not accepted. This sampling strategy was chosen to ensure participants were descendants, immigrants, and with a Danish origin, had different experience with loneliness, and were able to provide rich and relevant information related to the research question [36]. All participating students were contacted by email or telephone by the interviewer and received a detailed information letter about the study to give to their parents prior to interviews. Out of 19 contacted adolescents, four, all with ethnic minority background, declined participation without reason and 15 agreed to participate in the individual interviews (*n*_boys_ = 6, *n*_girls_ = 9). Of these, eight were born in Denmark to parents born in Denmark (Danish origin) and seven had an ethnic minority background, either born in Denmark (descendants) or abroad (immigrants) (Table 1).

### 2.2. Interview Guide and Interview Situation

The interview guide had a broad focus on how adolescents understand and experience loneliness and included themes inspired by the adolescent loneliness literature [36,37]. To gain insight into participants’ social lives and to build a confidant relationship [38], the first questions were related to everyday situations including mornings, school time, evenings, and weekends related to the participants’ friends, family, and interests. Participants were engaged with statements such as *“To begin with, please tell a little about yourself and your everyday life”* and *“Try to describe a typical day for me”*. Next, questions related to situations with happiness were articulated. This served as a platform for further talks about social relations and understanding of loneliness. We asked the participants to give examples of their personal experiences with loneliness and to clarify why they felt lonely with questions such as *”I would like to know more about loneliness; can you tell me what loneliness means to you? How would you define loneliness?”* and *“Can you please give examples on situations where you felt lonely? Why do you think these situations made you feel lonely?”*. The last questions in the interview guide were related to how to prevent and ease loneliness. Each interview ended with a debriefing.

The interviews were conducted in May and June 2012. They lasted between 22 and 61 min and were conducted after school in a classroom or a silent room provided by a teacher. Since loneliness is a sensitive subject, especially in adolescence [19], the interviewer attempted to create a confident room for conversations by emphasizing that loneliness is widespread in adolescence and by sharing personal experiences with loneliness. Two participants said they had never felt lonely. Yet, they still described their understandings of the phenomenon. Those who felt or had been feeling lonely gave detailed descriptions rooted in their own experiences. All interviews were completed in Danish, which most participants were fluent in. Only two participants had Danish as their second language and, thus, had a more limited vocabulary. The interviews were carried out without problems with good understanding and communication between the interviewer and the participants.

### 2.3. Data Analysis

All interviews were recorded with permission from the adolescents and transcribed verbatim by the interviewer. Participants and schools were assigned pseudonyms. Only K.R.M. and two coauthors (B.E.H. and T.T.-T.) read and discussed these paper transcripts, which were not shared with the participants. Each interview was read several times to validate content and meaning. The core of the analytical procedure was a thematic analysis [39], with some themes and categories predefined by the interview guide and others derived empirically. First, two main themes focusing on adolescents’ understandings of and experiences with loneliness were identified in the material. Second, our analysis focused on comparing thematic differences and similarities between adolescents with a Danish majority background and with an ethnic minority background (descendants and immigrants) within these two main themes. The main coding and analytical work was carried out by K.R.M. but discussed concurrently with B.E.H. and T.T.-T.

### 2.4. Ethical Issues

The study complies with national guidelines regarding ethical standards and data protection as they were in 2012 when the study was carried out, and it is registered at the Danish Data Protection Agency (case no. 2015-57-0008). There is no agency for ethical approval of qualitative studies in Denmark. Instead, the school principal and class teacher of each participant gave their approval of the study. The adolescents received an information letter addressed to their parents, giving them the opportunity to withdraw their child from the study. We informed the participants orally in class and prior to each interview that participation was voluntary, that they could withdraw any time, and that data would be anonymous.

Prior to the interviews, the interviewer told the informants that they were not obliged to talk about things they were not comfortable with [40]. Even though all adolescents expressed that loneliness is a sensitive and stigmatizing feeling, they indicated positive reactions after the interview, and some mentioned that it was a relief to talk to a stranger about how they really felt. In a few situations, major personal problems were articulated, and the interviewer took care of this by either contacting the school psychologist or by providing the adolescent with information on where to find help and meet like-minded young people.

## 3. Results

We derived two main themes related to how adolescents (1) understand and (2) experience loneliness. These are presented with subthemes individually below.

### 3.1. First Main Theme: The Characteristics of Loneliness

The first theme covered the adolescents’ understandings of loneliness as a phenomenon, where three subthemes emerged as distinct characteristics of loneliness: (1) loneliness as an adverse state, varying in intensity and duration, (2) loneliness as something different from being alone, and (3) loneliness as an invisible social stigma. We found no differences across ethnic backgrounds within this theme.

#### 3.1.1. Loneliness as an Adverse Feeling, Varying in Intensity and Duration

Loneliness was described as an emotional and adverse state associated with unpleasant feelings of emptiness, frustration, anger, and sadness. Several expressed that no one deserved to feel lonely and that they would feel sorry for others who felt lonely. Julia (D, school 1) defined loneliness as follows:


*Hmm. Emptiness (…) It is not like being alone; that you wish there was someone sitting next to you physically. It’s not a physical sensation, in a way. It is something you feel. (…) It is a negative feeling. For the most part, there is a difference between a negative and a positive feeling. You can feel it. And sometimes you can also be in doubt, but with the sense of loneliness you’re not in doubt.*


Thus, loneliness was emphasized as a feeling and a feeling that was generally negative. A few participants felt that loneliness was tolerable if it was not experienced too often and, eventually, a motive to seek company. Christian (DK, school 3) explained whether loneliness is negative or positive.


*I’d say mostly bad because with the experiences I have been through, to sit there alone and things like that. But sometimes, that loneliness may also be quite good because it may make you realize some things, realize that okay, nothing will come out of just sitting here.*


Loneliness was also described as a feeling varying in intensity and duration: some described themselves as feeling very lonely with experiences of severe, prolonged loneliness. Others described how they felt or had been feeling lonely in shorter or longer periods. These characteristics of loneliness as an adverse feeling that varied from transient to prolonged were common to all participants.

#### 3.1.2. The Distinction between Loneliness and Being Alone

The adolescents consistently distinguished between loneliness and being alone and some explained the distinction by emphasizing that loneliness was a feeling and being alone was more physical. Christian (DK, school 3) said:


*If I sit at home and I’m alone, then I will not feel lonely in that sense, because it is often something I have chosen and then it is because you’d like to just be yourself and just… either just sit and do nothing or sit and think some things through or homework and things like that.*


In a dialogue between the interviewer (I) and Julia (D, school 1), she explained how being alone can be both positive and negative.

J:
*You can definitely be alone without feeling lonely, e.g., if I have a very busy week and spend time with a lot of people, then I also need to be alone once in a while.*


I:
*Yes, and then being alone becomes a positive thing?*


J:
*Yes.*


I:
*When is being alone a bad thing?*


J:
*Hmmm, when… when you know that others do something that you’re not a part of.*


Being alone and loneliness was, however, described as related. Several participants gave examples of situations where they were physically alone—mostly at home—and felt lonely. Their experience of being alone seemed to depend on whether the situation was desired/voluntary or undesired/involuntary. Being among people was not considered a guarantee to avoid loneliness. Flora (DK, school 3) described how loneliness could also emerge in the presence of others.


*Because I can of course join a big group (and feel lonely), where you’re not really part of the conversation, or where you don’t feel like you’re part of the conversation, although the others, on the outside, see that you’re part of it, right.*


A few participants also emphasized how loneliness could be more intense when being among others, i.e., how the feeling was reinforced because it became more evident that one was not part of a group they wanted to be part of.

#### 3.1.3. Loneliness as an Invisible Social Stigma

A final important characteristic of loneliness was the social stigma it was related to. The feeling was described as hard to share because it is very, very difficult to talk about (Klara, DK, school 1) and something no one wanted others to know about. Most adolescents kept the feeling to themselves even though a few suggested that they could potentially talk to a very close friend or family member about it. The dialogue between Laura (D, school 5) and the interviewer illustrates how loneliness seems embarrassing.

L:
*If I felt lonely, I don’t think I would have told my friends.*


I:
*No, why do you think that is?*


L:
*I actually don’t know. But I don’t think I would have said it. Maybe I would have put it in another way; “I don’t think you call me or write to me as you usually do” and something like that.*


I:
*Why do you think loneliness is not something you talk about?*


L:
*That’s a good question. I think it’s embarrassing.*


As emphasized by Laura, using other words than “lonely” could be an alternative way to open a conversation about loneliness. Because of the hesitation to talk about loneliness, it became invisible. Several participants expressed how they thought no one else in their class felt lonely but them. They saw their classmates as happy and “fitting in”.

 *Uh, from class I would not think that any of the others are lonely. Everyone has like possibilities to hang out with someone. So do I. So it’s just whether you’re compatible with them on a like… psychological level. Whether you have the same way of thinking. Whether you have the same hobbies, the same things you like to do. Most in my class do, so I wouldn’t imagine anyone to be (lonely)… I think it’s because it (loneliness) is kept under wraps. I also don’t think that others would guess that I feel lonely.* (Mathias, DK, school 1)

Most of the adolescents expressed they were afraid of social consequences if their friends or classmates found out about their loneliness. They saw it as if their social status was at stake.

 *I just think people are trying to be a little more… in relation to the things people talk about, trying to be as little as possible something that is bad. Because you’re a little scared that otherwise they wouldn’t want to hang out with you or be friends with you.* (Julia, D, school 1).

Even though a few participants seemed a bit uncomfortable with the interview situation, the majority emphasized they had no problem talking to the interviewer about their loneliness because their social status was not at stake; the interviewer was a stranger who would disappear afterwards. Laura (DK, school 3) explained:


*I also think that because you’re a stranger, right, well I don’t know you at all, so I can just sit and say all sorts of things (…) I can say all the right things to you because no one will know, and you don’t know anyone to tell either.*


The idea of being the only one struggling with loneliness was described as intensifying the feeling. Klara (DK, school 1) explained how sharing loneliness with others could be a relief.


*I think it would help you in many ways to see that you’re not the only one! At least when I have it (feel lonely). Then, I often feel like the only one in class who feels that way. Or the only one at that time, who feels that way. Or like… you feel very alone with it (…). It’s always good to know that if there’s something you don’t like about yourself, or something you’re upset about, to know that you’re not the only one!*


The hesitation to share the feeling was dominant because of the fear of potential social consequences. Loneliness became an invisible stigma, and the adolescents seemed caught with an adverse feeling of loneliness, which intensified with the perception that they were the only one that was lonely.

### 3.2. Second Main Theme: The Perceived Causes of Loneliness

The second theme covered the adolescents’ experiences with loneliness. The richness of the descriptions of why the adolescents felt lonely gave considerable insight into numerous forms of perceived social deficiencies that emerged in the interrelation between the individual and their varying social contexts, e.g., at school, in the school class, in the broader community, or within the family. The social deficiencies were structured into those forms that appeared most strongly in the material, although they were overlapping and intertwined. Differences across ethnic backgrounds were few and primarily related to the adolescents’ individual characteristics. The extracts below from the interviews illustrate the variety, complexity, and depth of the individual experiences of loneliness.

#### 3.2.1. Lack of Feeling Close and Connected

The experience of lacking closeness and connection with friends or family was one common form of social deficiency described as a cause of loneliness. One boy explained how he felt lonely in class:

 *I like going to school. But I’m in some way—how do you say—lonely in the class. You see, I have friends in class, but they are not close friends. Not someone I bond with in class. Someone you’d jell with. Many of the girls have someone like that. Best friends or whatever it is that they have. I’ve never had that… It’s probably something to do with not having someone to go home with every day. There is not someone calling you regularly: “Do you want to go play football?” I’m not like the first person coming to mind if someone’s thinking, I’m bored, why don’t I hang out with someone. Or just the fact that no one reserves a seat for you in the bus if you’re going somewhere. Lonely in that way. Like someone is missing… someone who’s thinking about you. That’s what is missing.* (Mathias, DK, school 1)

Thus, Mathias described his loneliness as a lack of someone who remembers him and cares about him in his life, which was a form of social deficiency described by several of the adolescents. The extract also illustrates how one can have friends in class but still feel lonely because of a lack of closeness and connection. Mathias also explained how his loneliness was contextually dependent, and how he could forget about his loneliness when he was with good friends at a yearly summer camp. Despite these friends, who he communicated with on a regularly basis, he emphasized the lack of daily physical friendships as important to his loneliness.

#### 3.2.2. The Feeling of Not Being Accepted

The experience of not being accepted by others because one felt different, not fitting in, or feeling left out was the most common source to loneliness. Marie (DK, school 3) explained how her special interest in Asian pop music influenced the way her classmates thought about her and made her feel:


*I can also feel lonely because I have this interest in Kpop (pop music from Asia), where I think it’s hilarious and my friends are saying “what is this insane country you like watching?” And I’m thinking. Ah, like a punch in the face, right! (…) And then you can feel lonely when you don’t have any backup who accepts you for who you are.*


She emphasized the importance of finding someone who shares her interests and accepted her for these. The experience of not being accepted was also reported as an important source to loneliness by a boy, Abdul (I, school 4), who immigrated to Denmark from Turkey with his parents when he was 1 year old. He explained how he felt lonely when he started school because he was the only immigrant in his class. This experience made him feel different from the others and unaccepted, and he explained how he tried to be more like them to fit in and make them like him. This experience of not feeling accepted emerged in several interviews. Moreover, Sofia (I, school 4), explained how she sometimes felt lonely in her school class because she was Polish, and the majority of her classmates were Muslim who often spoke Turkish and fasted during Ramadan.

A few participants gave examples of how a change of school had relieved their loneliness. The following extract from Julia (D, school 1) illustrates the change of context:


*Before, I was in a class with only Danes. Everyone had blue eyes and was fair. And there was no one like Claudia (classmate who is half Austrian), no one that was half something else too, which there are so many of here (in her new class). But they were all entirely Danish, and I believe they think it was strange that I was from another country, and I was sometimes also teased with it, and they would never do that here… And then I came to this school, and I was a completely different person, and I felt accepted.*


The quote illustrates the importance of the school class, and how it holds opportunities for engaging in social relations. It can provide you with someone like you, who shares interests, music, religion, or other important features, and it can provide a positive and accepting school class climate that accepts individuals as they are. The quote also elucidates how loneliness may ease or disappear when you meet someone who is “like you” or accepts you as you are. Acceptance and belongingness were related to important aspects of daily lives, e.g., hobbies, ethnic background, religion, preference for music, or appearance. The quotes also highlight that loneliness emerges through a combination of characteristics of the individual and the social context. This analytical point clarifies how minority status—related to, e.g., ethnicity or hobbies—is a contextual phenomenon and that it is not just the minority status per se that induce feelings of loneliness.

#### 3.2.3. Feeling Alone with Problems

The feeling of being alone with a major concern or problem was a third common course of loneliness. The feeling was described as a consequence of not being able to express a concern or a problem to others. Flora (DK, school 3) said:


*And then when you’re alone, then… you’re like completely empty inside and then those problems emerge and things like that, right. Then you feel very lonely because you can’t tell anyone.*


Even though these adolescents felt lonely, they still talked about close and accepting relationships with few or certain groups of friends or family. One example was given by Nikolaj (D, school 2), who was born in Denmark to a father from Bosnia and a mother from Montenegro. He spoke about his father’s depression and violent behavior as a consequence of the war in Bosnia. His father’s mental condition influenced the family, and Nikolaj spoke about domestic violence. He felt lonely, because he felt alone with this family matter, even though he had good friends in class.


*I don’t think my sister takes a lot of responsibility (in relation to problems in his family). When they (his parents) fight and all that… maybe she will interrupt. But she will not do anything special to stop it. She just wants to sit and shout at them and things like that… you see… then I feel like I’m alone with this… And I guess that means that you have a lot to go through on your own.*


What seemed to be vital for feeling lonely was the lack of emotional expression and possibilities to share concerns.

Sophia (I, school 4) described how talking to her parents about problems at school could ease her loneliness:


*I do feel lonely sometimes… well because if you’re like lonely sometimes, it can also be good, because you think about things, and then afterward I go to my parents and talk to them about problems in school. We just talk and then… then I feel better.*


This type of social deficiency was experienced by adolescents across ethnic backgrounds.

#### 3.2.4. Lack of Feeling Understood

Another common form of social deficiency described as a cause of loneliness was not feeling understood by others. Sophia (I, school 4) emigrated from Poland with her parents when she was 10 and described how she could feel lonely when she did not feel understood:


*Hmm, yes, it can also be your background; I’m not Danish—or that I come from Poland or that I’m not Muslim or that I’m a Jehovah’s witness—it can also be that kind of thing. That I explain that I’m not joining Christmas things or that I don’t celebrate birthdays and things like that. And they just tell me how much fun it is. And then they keep saying, just come to the party, nothing will happen. But I don’t feel like they understand me.*


Thus, Sofia felt her classmates did not understand her choices, which were based on her religion. She also described how a recent school change made a difference because her new teachers understood her religion and background.

Language proficiency was another barrier Sophia mentioned where she found it difficult to argue for norms and values that were different from her classmates. Sofia explained how she would probably feel less lonely if she was able to explain herself properly:


*Maybe I should improve my Danish. Then, I can also give them (classmates) more arguments for why I don’t go to birthdays and things like that…*


The experience of not feeling understood as a cause of loneliness was also described by Adila (I, school 4).

A:
*It’s just that I can’t talk to anyone about my problems. About what’s happening to me… about things that have changed.*


I:
*Yes, what makes you think you can’t talk to anyone about it?*


A:
*They won’t really understand me. They’re like… they haven’t been through the problems I have, so they take it in another way. They see it in another way, so…*


I:
*Yes, so it (loneliness) is also about feeling understood?*


A:
*Yes*


I:
*Are you thinking about your family or more about those from school when you think about them?*


A:
*Well, both. Both those from school and my family.*


Thus, Adila described how friends and family are not able to relate to and understand her problems. Both Adila and Sofia still felt lonely despite close and supportive family or friends in class or Kingdom Hall. What seemed to be vital for their loneliness was the experience of not feeling understood by friends, family, or teachers in their everyday life. Even though both girls were immigrants, only Sofia’s loneliness was related to her ethnic background as she did not feel understood by friends and teachers at school about her religion, cultural norms, and language.

#### 3.2.5. Feeling Neglected

The fifth common form of social deficiency described as a cause of loneliness was the experience of feeling neglected, rejected, or overlooked by others. Flora (DK, school 3) described a period with intense loneliness:


*It began with a lot of stuff in my family, because we, this sounds pretty lame right, but we just had a baby brother and we were building an extension and then we had very little money and everything (…) The only time there was left was for my baby brother, because he was of course so small. And then I started to feel a little lonely at home, right… Lonely at home and take care of everything yourself. My parents were always upset because they didn’t have the energy for anything, and then I also got easily upset.*


The parents’ absence made Flora feel neglected and lonely. In contrast, Abdul (I, school 4), who was not feeling lonely at the time of the interview, explained that his family was very concerned about him. Thus, having a family that is present and acknowledges one’s importance seemed crucial to avoid loneliness.

Several participants also described how they would feel lonely if their friends would not hang out with them or if they felt rejected. Julia (D, school 1) elaborated on a situation in which she had been feeling lonely:

J:
*That I was forced to be somewhere I didn’t feel comfortable, and there was no one there I knew. AND that I knew that the other girls from class were together somewhere else, where I wasn’t with them.*


I:
*Yes, because your mother said you had to go to the club?*


J:
*No, I could go with them! They just didn’t want me to go! They went in a group, and other girls from class couldn’t come, and they were like a group of five or so, and the rest couldn’t join.*


Thus, the feeling of being neglected and left out by friends was an important social deficiency leading to feelings of loneliness. The significance of feeling included by peers was also described by Christian (DK, school 3).

C:
*I also don’t think it would help me if someone came over and said: but why are you sitting there all alone. Because it may seem a little like, yeah, well done pointing out the most obvious! But perhaps coming over to ask, why don’t you join us? (…) In a way that you would feel somewhat included. Instead of me having to be the one to make sure I’m included. Because then you feel like you’re just a third wheel, imposing yourself on all the others.*


I:
*So, it’s important that you’re invited to join the group instead of having to…*


C:
*Yes, of course you have to do something yourself sometimes. Of course, you do, but if it’s you having to do it every time, then it becomes really difficult.*


This extract highlights the importance of feeling accepted and included by others, and, furthermore, that the context also possesses a responsibility to be including as it may be difficult to invite yourself into the company of others.

## 4. Discussion

This paper explored how adolescents understand and experience loneliness in their everyday social lives with a special focus on the role of their migration status. The findings highlighted an important aspect that has received little attention in the previous literature on migration and loneliness: the importance of the social context that the individual is embedded in. Thus, this study contributes to the current literature with the finding underlining that specific individual characteristics are important for experiencing loneliness only because they are embedded interact with the social contexts the individual is embedded in. Another new finding is that we observed more similarities than differences among adolescents of different migration status. Generally, loneliness was described as an adverse feeling, distinct from being alone. The feeling varied in intensity and duration and was associated with feelings of sadness, emptiness, and anger. There was an overriding hesitation to share feelings of loneliness with others because of a fear of the potential social consequences. Thus, loneliness became an invisible social stigma where many seemed trapped in the experience of being the only lonely. A variety of perceived social deficiencies, e.g., the feeling of not being accepted, a lack of closeness and connection, and a feeling of not being understood, were emphasized as causing loneliness and emerged in the interrelation between the individual and their social context. The results suggest that it is not the presence of specific individual characteristics such as those related to having a particular migration status, type of interest, concern, or problem that causes loneliness; their importance for loneliness was dependent on the social context the individual found themselves in. Differences across migration status became apparent in the individual characteristics, where some were related to the adolescents’ ethnic minority background, for example, language barriers, being the only immigrant in class, domestic violence, and experiences of not being understood in terms of one’s cultural norms and values. The findings in this study highlight the importance of combining characteristics of the individual and the social context when studying antecedents to adolescents’ loneliness.

The finding that loneliness was characterized as an adverse feeling, varying from short to prolonged and with a reference to distressing emotions, is supported by a few other empirical studies on adolescent loneliness [16,19,20,21,23]. These studies were carried out in several countries, which supports our finding that these characteristics of loneliness are more general characteristics of the phenomenon and not conditional on one’s ethnicity. Our results also suggest that loneliness and being alone are understood as two conceptually distinct concepts. This is in accordance with previous qualitative studies of children and adolescents [20,21,22,23,41] and in agreement with widely acknowledged conceptualizations of loneliness [16,42]. For example, the study by Martin and colleagues [22] described loneliness with negative associations and coupled it with sadness and hopelessness. In contrast, aloneness had both positive and negative associations that were influenced by the setting and the frequency of being alone. The stigmatizing nature of loneliness is also emphasized by other researchers [19,43,44], with recent work showing much the same thing [45]. Yet, the theme was not emphasized in previous qualitative studies among 8–13-year-olds [21,22,23] and, therefore, represents new insight into loneliness in adolescence. One reason could be that our study was carried out among a slightly older population. According to Parkhurst and Hopmeyer [19], adolescence is a period in life in which the awareness of one’s social standing within the peer group increases, and where shame over lack of social competences dominates.

In agreement with our study, some other researchers found that loneliness is an emotional response related to deficiencies in social relations [16,21,23,46]. For example, Chipuer [21] found that most of the participating children explained their loneliness by social deficits such as having no one to play with, being rejected by peers, and having no friends. Hymel et al. [20] found loneliness to be a multidimensional phenomenon with a cognitive dimension. Central to this dimension was the perception that relational provisions such as inclusion, affection, and reliable alliances were not being met. Our study also suggests that perceived social deficiencies emerge in the interrelation between the individual and their social context. In accordance with a few other researchers, this finding highlights that loneliness in childhood and adolescence is about the individual’s interrelations [22,24,47,48]. Korkiamäki [24] concluded that the social, spatial, and cultural setting adolescents live within influences their opportunities to make friends. Furthermore, Dahlberg [48] emphasized that the experience of loneliness is closely related to its context: “*loneliness cannot be understood in another way than in relation to our existence with others*” (p. 204). It is, therefore, important to include both characteristics of the individual and social contexts when studying adolescents’ loneliness. We found the individual and social contexts leading to loneliness to vary greatly between adolescents, and to be rooted in their everyday social lives. Several other researchers suggest that the life conditions in which one is embedded, e.g., one’s history and social background, shape the person’s experience and definition of loneliness [32,49].

The perceived causes of loneliness were not fundamentally different across migration status. Our findings support the importance of the individual factors regarding loneliness among immigrant adolescents or descendants that have already been addressed in the literature. However, our study highlights the importance of context in the experience of loneliness during adolescence. That is an important discovery that has not been discussed in the literature on migration and loneliness. Such a finding has important implications for future interventions to reduce or prevent loneliness among adolescents, and it highlights that the context is as important to address in interventions as individual characteristics.

### 4.1. Methodological Reflections and Future Implications

It is a strength in this study that we applied maximum variation in the sampling strategy of schools, which resulted in a study population with mixed migration status, involving schools with low, middle, and high proportions of ethnic minorities (immigrants and descendants). Including more adolescents with diverse migration status might have provided the study with more profound and varied findings including more varied individual causes of loneliness related to, e.g., migration status. However, the overall finding of how the context is just as important as the individual characteristics as antecedents to loneliness would more than likely remain.

Loneliness is a sensitive subject to talk about, especially in adolescence [19]. It is possible that the interviews suffer from social desirability bias because some of the participants may have withheld or underestimated their experiences with loneliness in the interviews. The interviewer tried to overcome this potential bias by emphasizing loneliness a widespread phenomenon in adolescence and by sharing personal experiences with loneliness. Even though some of the adolescents seemed reluctant in the interview situation, most expressed that the interview had been pleasant and that it had been a relief to talk to a stranger about their loneliness. The careful building of confident relationships with the participants resulted in candid interviews.

When a study involves a minority group, it is important to consider how the research portrays this group and how it affects it. To avoid negative stereotyping of ethnic minority groups [33], we focused on both similarities and differences among the three groups of adolescents: Danish majority and adolescents with an ethnic minority background (immigrants and descendants). Furthermore, conducting research among adolescents about a sensitive subject such as loneliness may have other ethical implications that need to be considered: Does it increase the problem by making adolescents reflect unnecessarily upon their social relations and loneliness? Does it contribute to easing loneliness by talking about the feeling and addressing it as prevalent? In our interviews, several of the participants expressed that the feeling of being the only one that was lonely made the feeling even stronger. Fortunately, many expressed that they felt relieved talking about their loneliness.

### 4.2. Directions for Future Research

This study is, to our knowledge, the first qualitative study to explore how adolescents understand and experience loneliness across migration status, and it is, therefore, important to carry out additional research to support or challenge our findings.

Our findings suggest that the social context—and especially the school class—plays an important role in adolescent loneliness; school is a place to be accepted or included, and it can provide opportunities to share interests, norms, values, etc. An important theme for future research is, therefore, to look more closely into how the school class context may be related to adolescents’ loneliness, e.g., by examining whether loneliness is connected to the composition of individuals within a school class or the school class climate.

### 4.3. Implications for Practice

Among immigrants, their descendants, and Danish native-born majority adolescents, there are large numbers of adolescents who experience intense or chronic feelings of loneliness, and the prevalence may be increasing [12,50,51]. It is, therefore, important to address how to reduce loneliness across all groups. The public school system comprises mostly children and adolescents, making it a unique and important arena for interventions aimed at reducing and preventing loneliness in adolescence [52,53]. The fact that loneliness increases the risk for health-related, behavioral, and academic problems among adolescents emphasizes the importance of early identification and prevention programs. Therefore, effective programs to prevent adolescents’ loneliness are important [53]. A meta-analysis of interventions for youth to reduce loneliness has shown that improving social skills, enhancing social support, increasing opportunities for social contact, and addressing maladaptive social cognition can reduce adolescent feelings of loneliness [54]. Most of the studies in this meta-analysis focused on children and adolescents with special mental or physical needs or adults and older people. Intervention research that focuses on how to prevent or reduce loneliness in adolescents without special needs is still sparse [55]. Qualter and colleagues [15] suggested interventions focusing on attention reprogramming and retraining negative thoughts to be the most appropriate interventions programs to alleviate prolonged loneliness in adolescence. The findings in this paper suggest that a focus on the social context such as the school class may be an additional important component to address in future intervention programs, and this is supported by findings of the importance of context in this paper. A focus on changing the school class climate to be more including and accepting of differences [56] may have positive implications for loneliness and may be particularly feasible in public schools that comprise the most adolescents.

## 5. Conclusions

The new contribution of this study is that specific individual characteristics are important for understanding loneliness only because they interact with the social contexts the individual is embedded in. Differences across migration status were few and mostly related to variations in the adolescents’ individual characteristics. Specifically, our findings highlight the importance of (1) studying characteristics of both the individual and the social context in research on the antecedents to adolescents’ loneliness, and (2) applying this perspective in other studies on the importance of migration status.

## Figures and Tables

**Table 1 ijerph-18-11425-t001:** Characteristics of participants.

Name ^1^	School	Age at Interview	Gender	Current or Former Experience with Loneliness	Country of Birth of Interviewee and Their Parents ^2^
Klara	1	14	Girl	Yes	Born in Denmark to parents born in Denmark (DK)
Mathias	1	15	Boy	Yes	Born in Denmark to parents born in Denmark (DK)
Julia	1	14	Girl	Yes	Born in Denmark, mother born in Denmark and father born in Italy (D)
Nikolaj	2	14	Boy	Yes	Born in Denmark, mother born in Montenegro and father born in Bosnia (D)
Alexandra	2	14	Girl	No	Born in Denmark to parents born in Denmark (DK)
Marie	3	15	Girl	Yes	Born in Denmark to parents born in Denmark (DK)
Christian	3	15	Boy	Yes	Born in Denmark to parents born in Denmark (DK)
Flora	3	15	Girl	Yes	Born in Denmark to parents born in Denmark (DK)
Sofia	4	14	Girl	Yes	Born in Poland to parents born in Poland (I)
Abdul	4	15	Boy	Yes	Born in Turkey to parents born in Turkey (I)
Adila	4	15	Girl	Yes	Born in Iraq to parents born in Iraq (I)
Ramtin	4	14	Boy	Yes	Born in Denmark to parents born in Turkey (D)
Simon	4	14	Boy	No	Born in Denmark to parents born in Denmark (DK)
Laura	5	14	Girl	Yes	Born in Denmark to parents born in Eritrea (D)
Christina	5	14	Girl	Yes	Born in Denmark to parents born in Denmark (DK)

^1^ Pseudonyms; ^2^ in the results section, the following abbreviations are used: (DK), Danish origin; (D), descendant; (I), immigrant.

## Data Availability

Funding contracts mean that data underlying this article are not publicly available. Please contact the first author to gain access to the data.

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
