# Peer review of "Lonely, but Not Alone: Qualitative Study among Immigrant and Native-Born Adolescents"

_ijerph, 2021, doi:10.3390/ijerph182111425_

Round 1

Reviewer 1 Report

A well-written paper that adds to the current body of literature. A

pleasure to read. I look forward to reading of the outcomes for this current

project. I would really like it that the sample increase in future studies to university students

Author Response

  1. A well-written paper that adds to the current body of literature. A pleasure to read. I look forward to reading of the outcomes for this current project. I would really like it that the sample increase in future studies to university students

Response: Thank you very much for your positive review of the manuscript. This is highly appreciated.

Reviewer 2 Report

I can’t find the specific contribution of this study to the knowledge of loneliness in adolescence. Previous studies, both qualitative and quantitative, have shown very similar results and conclusions. So, what is the novelty reported? It must be clearly identified, both in the Introduction and in the Discussion section. Unfortunately, I have the impression that results are in line with previous literature, and the contribution of the present research is very limited.

To my view, the design of the study makes quite difficult to find differences between participants due to their ethnicity. The number of participants as well as the lack of specific questions about cultural loneliness is important.

One way to solve this limitation could be to include an additional study with a design that helps to identify the specific role of ethnicity in the perception and experience of loneliness for adolescents.

Reviewer 3 Report

Thank you for the opportunity to review this manuscript. Overall, the quality of writing within this paper on loneliness among youth of diverse ethnic backgrounds is strong, as well as there is a lot of interesting information presented. I do have suggestions for refinement that will further strengthen this work.

  1. I recommend the authors review COREQ criteria by Tong et al. for reporting of qualitative studies and incorporate these throughout.
  2. The methodological design is missing completely from this manuscript. Conducting interviews is not a methodological design, but rather a data collection approach. This needs to be included in the abstract and explicitly outlined in the materials and methods section. It wasn’t until page 12 under 4.1 that maximum variation sampling was mentioned.
  3. There is no mention of recruitment processes. How were adolescents within the selected schools identified? What steps were taken for informed consent processes? How were parents involved? It sounds like data were collected from the adolescents regarding loneliness before agreeing to participate (from the entire class). Please clarify. It isn’t until later that it was noted ethics board approval is not needed for qualitative studies in Denmark. Given this is a sensitive topic and it was conducted with minors, it is quite troubling to think there was no ethical oversight and that parental informed consent and adolescent assent were not conducted prior to enrolling the participants. This is especially true given the authors note a few situations where major personal problems were identified during the study.
  4. Why were the interviews from the one school with only 2 participants excluded from the analysis?
  5. Why were focus groups conducted with all participants followed by individual interviews? What was the rationale for this approach? The interview guide is mentioned; however, it isn’t clear how the focus group interview guide differed from the individual interviews. Were questions asked specific to their ethnic diversity? Very little information was shared related to this in the results.
  6. What steps were in place to protect adolescents? Since the authors clearly and appropriately indicate the sensitive nature of loneliness, what was the justification for openly discussing this topic in a group forum potentially stigmatizing participants or causing emotional distress?
  7. Fluency in Dutch was noted in all participants except for two. What language were interviews and focus group discussions conducted in? If they were conducted in Dutch, what processes were in implemented to ensure accurate translation of the interviews, as the results (i.e., quotes) are presented in English.
  8. As thematic analysis is identified on page 4 and noted to be partially derived from the interview questions, it would be helpful to include the actual interview guides. Who did the analysis? Was software used or not? What processes were in place to ensure rigor of analytical procedures? There is insufficient detail in the analysis/methods to have confidence in the process. This likely can be rectified through added details.
  9. Demographics on participants included should be presented in the results section.
  10. What was the racial diversity within the sample, if any?
  11. Were similarities noted between adolescents from diverse ethnicities who participated or in comparison to non-ethnically diverse adolescents? This was not made explicitly clear.

Round 2

Reviewer 2 Report

Agree with changes

Reviewer 3 Report

Thank you for your attention to the suggested changes.